# SEMANTIC DECOUPLED DISTILLATION

## ABSTRACT

Logit knowledge distillation attracts increasing attention due to its practicality in recent studies. This paper argues that existing logit-based methods may be sub-optimal since they only leverage the global logit output coupled with multiple semantic knowledge. To this end, we propose a simple but effective method, i.e., semantic decoupled distillation (SDD), for logit knowledge distillation. SDD decouples the logit output as multiple local outputs and establishes the transferring pipeline for them. This helps the student to mine and inherit richer and unambiguous logit knowledge. Besides, the decoupled knowledge can be further divided into consistent and complementary logit knowledge that transfers the multi-scale information and sample ambiguity, respectively. SDD introduces dynamic weights for them to adapt to different tasks and data scenes. Extensive experiments on several benchmark datasets demonstrate the effectiveness of SDD for wide teacher-student pairs, especially in the fine-grained classification task.

## 1 INTRODUCTION

Knowledge distillation is a general technique for assisting the training of "student" networks via the knowledge of pre-trained "teacher" networks (Gou et al., 2021). Depending on the location of transferred knowledge, distillation methods are divided into two categories: logit-based distillation (Hinton et al., 2015) and feature-based distillation (Romero et al., 2014). Due to the computational efficiency (Zhao et al., 2022) and ability to handle heterogeneous knowledge (Wei et al., 2023), logit distillation has gained increasing attention in recent years.

To date, many logit distillation methods have been proposed and they can be broadly categorized into two groups. The first group aims to extract richer logit knowledge by introducing multiple classifiers (Yao & Sun, 2020; Zhu et al., 2018) or self-supervision learning (Xu et al., 2020a). The second group aims to optimize the knowledge transfer by techniques like dynamic temperature (Xu et al., 2020b) or knowledge decoupling (Zhao et al., 2022; Yang et al., 2023; Xue et al., 2021). While these methods achieve good results, we argue they would lead to sub-optimal results since they only leverage the global logit knowledge of the whole input.

Specifically, the whole image usually couples the information of multiple classes. As shown in the first column in figure 1(a), two classes, like classes 5 and 6, may belong to a superclass, like fish, and their samples have similar global information, such as shape. Besides, as shown in the second column in figure 1(a), one scene may contain the information of multiple classes, like classes 6 and 983, resulting in a semantically mixed logit output. Consequently, multiple distinct and fine-grained semantic knowledge is coupled in a single logit output. This would hinder the student from inheriting comprehensive knowledge from the teacher, resulting in sub-optimal performance.

To this end, we propose the SDD method to assist the logit distillation by decoupling the logit output at the scale level. Specifically, as shown in Fig. 1(b), SDD decouples the logit output of the whole input into the logit outputs of multiple local regions. This helps acquire richer and unambiguous semantics knowledge. Then SDD further divides the decoupled logit outputs into consistent and complementary terms according to their class. Consistent terms belong to the same class with the global logit output, which transfers the multi-scale knowledge of corresponding classes to students. Complementary terms belong to different classes from the global logit output, which preserves the sample ambiguity for the student, avoiding overfitting the ambiguous samples. Moreover, SDD introduces dynamic weights for them to adapt to different tasks and data scenes. Finally, SDD

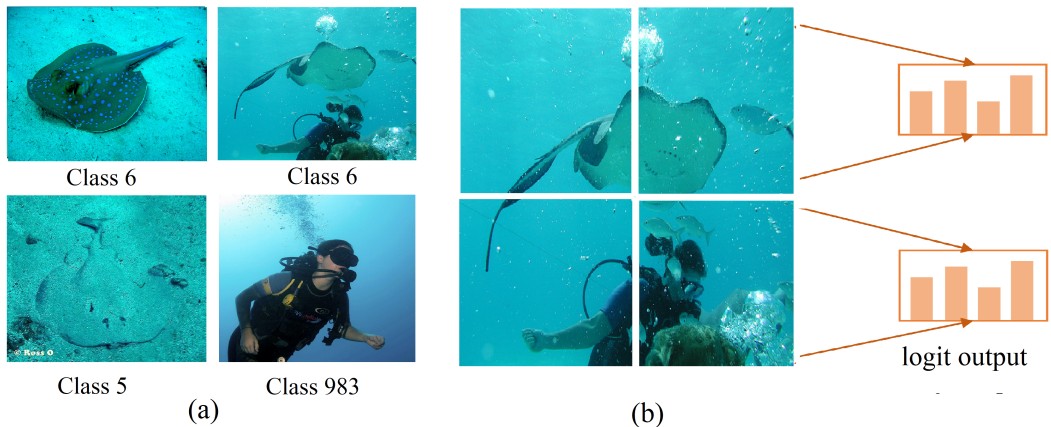

Figure 1: Image visualization on ImageNet. the top line of (a) shows some misclassified samples in ResNet34. The bottom line of (a) shows the corresponding wrong class and samples. (b) illustrates the intuitive model for semantic decoupling.

performs the distillation among all these logit outputs to transfer comprehensive knowledge from the teacher to the student, improving its ability to perceive local information.

In total, we summarize our contributions and the differences from the existing approaches as follows:

- We reveal the weakness of the classic logit distillation caused by the coupling of multi-class knowledge. This limits the ability of students to inherit accurate semantic information for ambiguous samples.

- We propose a simple but effective method, i.e., SDD, for logit knowledge distillation. SDD decouples the global logit output as consistent and complementary local logit output and establishes the distillation pipeline for them to mine and transfer richer and unambiguous semantic knowledge.

- Extensive experiments on several benchmark datasets demonstrate the effectiveness of SDD for wide teacher-student pairs, especially in the fine-grained classification task.

## 2 RELATED WORK

**Feature-based Distillation**. The feature-based distillation is first proposed in Fitnets (Romero et al., 2014), in which the student is trained to mimic the output of the intermediate feature map of the teacher directly. Since then, a variety of other methods have been proposed to extend the fitnets by matching the features indirectly. For example, AT (Komodakis & Zagoruyko, 2017) distills the attention map of the sample feature from a teacher to a student. Besides, some methods are further proposed to transfer the inter-sample relation, such as RKD (Park et al., 2019), SP (Tung & Mori, 2019), and reviewKD (Chen et al., 2021b). While these methods achieve good results, their ability to handle heterogeneous architecture is limited (Passalis, 2020; Chen et al., 2021a). Therefore, this paper pays attention to the logit-based distillation that has good generalization for heterogeneous knowledge distillation

**Logit-based Distillation**. The logit-based distillation is originally proposed by Hinton (Hinton et al., 2015), in which the student is trained to mimic the softened logit output of the teacher. Compared to one-hot labels, the soft logit output provides additional information of the inter-class similarity. To better measure this similarity and reduce the impact of noise in one-hot labels, FN (Xu et al., 2020b) introduces the L2-norm of the feature as the sample-specific correction factor to replace the unified temperature of KD. SSKD trains extra classifiers via the self-supervision task to extract "richer dark knowledge" from the pre-trained teacher model. KDExplainer (Xue et al., 2021) proposes a virtual attention module to improve the logit distillation by coordinating the knowledge conflicts for discriminating different categories (Xu et al., 2020a). SSRL transfers the knowledge by guiding the

teacher's and student's features to produce the same output when they pass through the teacher's pre-trained and frozen classifier (Yang et al., 2020). SimKD transfers the knowledge by reusing the teacher's classifier for the student network (Chen et al., 2022). In addition, the latest research proposes the decoupled knowledge distillation that divides the logit knowledge into target knowledge and non-target knowledge (Zhao et al., 2022). NKD further proposed to normalize the non-target logits to equalize their sum, transferring the teacher's non-target knowledge (Yang et al., 2023). However, all of them only focus on the global logit knowledge with the mixed semantic, hindering the student from inheriting the comprehensive knowledge from the teacher.

## 3  METHOD

In this section, we revisit the conventional KD and then describe the details of our proposed semantic knowledge distillation.

**Notation.** Given an image input $x$, let $T$ and $S$ denote the teacher and student networks, respectively. We split these networks into two parts: (i) one is the convolutional feature extractor $f_{Net}, Net = \{T, S\}$, then the feature maps in the penultimate layer are denoted as $f_{Net}(x) \in R^{c_{Net} \times h_{Net} \times w_{Net}}$, where $c_{Net}$ is the number of feature channels, $h_{Net}$ and $w_{Net}$ are spatial dimensions. (ii) Another is the projection matrix $W_{Net} \in R^{C_{Net} \times K}$, which project the feature vector extracted from $f_{Net}(x)$ into $K$ class logits $z_{Net}^l$, $l = 1, 2, ..., K$. Then, let $f_{Net}(j, k) = f_{Net}(x)(:, j, k) \in R^{c_{Net} \times 1 \times 1}$ denotes the feature vector at the location $(i, j)$ of the $f_{Net}(x)$. According to the analysis in (He et al., 2015; Ren et al., 2015), $f_{Net}(j, k)$ can be regarded as the representation of the region $(t_x, t_y, t_x + d, t_y + d)$ in $x$, where $t_x = d * j$, $t_y = d * k$ and $d$ is the downsampling factor between the input and the final feature map.

### 3.1  CONVENTIONAL KNOWLEDGE DISTILLATION

The concept of knowledge distillation was first proposed in (Hinton et al., 2015) to distill the logit knowledge from the teacher to the student by the following loss,

$$
\begin{cases}
L_{\mathcal{KD}} = \mathcal{KL}(\sigma(P_T) \| \sigma(P_S)) & (1) \\[2mm]
P_t = W_T \sum_{j=0}^{h_T-1} \sum_{k=0}^{w_T-1} \frac{1}{h_T w_T} f_T(j, k) & (2) \\[2mm]
P_s = W_S \sum_{j=0}^{h_S-1} \sum_{k=0}^{w_S-1} \frac{1}{h_S w_S} f_S(j, k) & (3)
\end{cases}
$$

where $\sigma(.)$ is the softmax function and $\mathcal{KL}(., .)$ means the KL divergence.

Due to the linearity of fully connected layer, $P_T$ and $P_S$ can be rewrite as follows,

$$
\begin{cases}
P_T = \sum_{j=0}^{h_T-1} \sum_{k=0}^{w_T-1} \frac{1}{h_T w_T} W_T f_T(j, k) = \sum_{j=0}^{h_T-1} \sum_{k=0}^{w_T-1} \frac{1}{h_T w_T} L_{Tjk} & (4) \\[2mm]
P_S = \sum_{j=0}^{h_S-1} \sum_{k=0}^{w_S-1} \frac{1}{h_S w_S} W_S f_S(j, k) = \sum_{j=0}^{h_S-1} \sum_{k=0}^{w_S-1} \frac{1}{h_S w_S} L_{Sjk} & (5)
\end{cases}
$$

where $L_T$ and $L_S$ means the logit output maps of teacher and student, respectively.

From the above equations, we can see that the conventional logit-based distillation only leverages the average logit output that mixes different local logit knowledge calculated from different local feature vectors, such as $L_{T11}$. However, as shown in figure 1, different local outputs usually contain distinct semantic information. Simply fusing them in a logit output would hinder the student from inheriting the comprehensive knowledge from the teacher. Consequently, conventional logit-based distillation usually leads to sub-optimal performance.

To overcome this limitation, we propose the SDD that decouples the logit output at the scale level to mine richer and unambiguous logit knowledge for student learning.

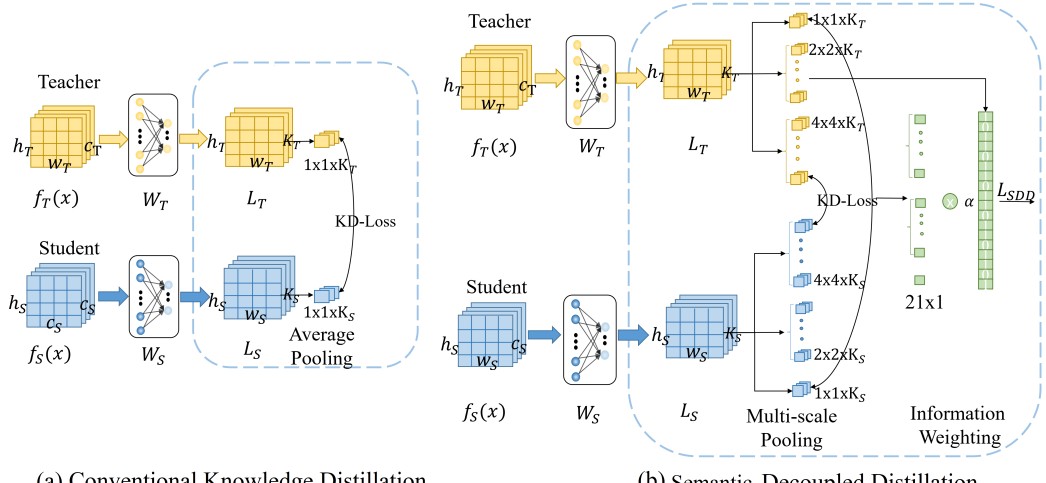

(a) Conventional Knowledge Distillation       (b) Semantic Decoupled Distillation

Figure 2: Illustration of the conventional KD (a) and our **SDD (b)**. Compared with the conventional KD that only considers the global logit knowledge via global average pooling, SDD proposes to capture the multi-scale logit knowledge via the multi-scale pooling so that the student can inherit the fine-grained and unambiguous semantic knowledge from the teacher.

## 3.2 SEMANTIC DECOUPLED KNOWLEDGE DISTILLATION

In this section, we describe the proposed SDD in detail. As shown in Fig. 2(b), the SDD consists of two parts: multi-scale pooling, and information weighting. Specifically, given the logit output maps of teacher and student, i.e. $L_T$ and $L_S$ via Eq. 5 and Eq. 5, multi-scale pooling runs the average pooling on different scales to acquire the logit output of different regions of the input image. Compared with the conventional KD that only considers the logit knowledge via global average pooling, this helps preserve more comprehensive teacher knowledge for the student, specifically the fine-grained semantic knowledge from the teacher. Then, we establish the knowledge distillation pipeline for the logit output of each considered region. Finally, information weighting adjusts the weight of the distillation loss for the local logit that has inconsistent classes with the global logit. This guides the student network to pay more attention to the ambiguous samples whose local and global categories are inconsistent.

Specifically, the multi-scale pooling splits the logit output map into cells at different scales and performs average pooling operations to aggregate the logit knowledge in each cell. Let $\mathcal{C}(m,n)$ denotes the spatial bins of the $n_{th}$ cell at $m_{th}$ scale, $\mathcal{Z}(m,n)$ denotes the input region corresponding to this cell, $\pi_T(m,n) \in R^{K_T \times 1 \times 1}$ denotes the logit output of the teacher for region $\mathcal{Z}(m,n)$, which is the aggregated logit knowledge of this cell,

$$\pi_T(m,n) = \sum_{i,j \in \mathcal{C}(m,n)} \frac{1}{m^2} L_T(j,k) \tag{6}$$

where $(i,j)$ means the coordinate of the logit output in $\mathcal{C}(m,n)$. And the paired logit output of the student for the same region $\mathcal{Z}(m,n)$ is the $\pi_S(m,n) \in R^{K_S \times 1 \times 1}$,

$$\pi_S(m,n) = \sum_{i,j \in \mathcal{C}(m,n)} \frac{1}{m^2} L_S(j,k) \tag{7}$$

where $m$ and $n$ are as same as those in $\pi_T(m,n)$. For each paired logit output, the distillation loss $\mathcal{D}(m,n)$ that transfers the logit knowledge at the region $\mathcal{Z}(m,n)$ from the teacher to the student is defined as follows,

$$\mathcal{D}_{(}m,n) = \mathcal{LD}(\sigma(\pi_T(m,n)), \sigma(\pi_S(m,n))) \tag{8}$$

where $\mathcal{LD}(.,.)$ denotes the conventional logit-based distillation loss, such as the KL divergence in (Hinton et al., 2015) and the decoupling loss in (Zhao et al., 2022). By traversing all the scales $m$ in $M = \{1, 2, 4, ..., w\}$ and their corresponding cells $N_m = \{1, 4, 16, ..., w^2\}$, we can get the final SDD loss as follows,

$$\mathcal{L}_{SDD} = \sum_{m \in M} \sum_{n \in N_m} \mathcal{D}(m, n) \tag{9}$$

Besides, we can further divide the decoupled logit outputs into two groups via their classes. One is the consistent terms that belong to the same class with the global logit output. Another is the complementary terms that belong to different classes from the global logit output. Here, the consistent terms transfer the multi-scale knowledge of corresponding classes to students. The complementary terms preserve the sample ambiguity for the student, avoiding overfitting the ambiguous samples. help regularize the learning of the student. Specifically, when the global prediction is right while the local prediction is wrong, the inconsistent local knowledge encourages the student to preserve the sample ambiguity, avoiding overfitting the ambiguous samples. On the other hand, when the global prediction is wrong while the local prediction is right, the inconsistent local knowledge can encourage the student to learn from similar components among different categories, alleviating the bias caused by the teacher.

Here, we introduce dynamic weights for them to adapt to different tasks and data scenes. And $L_{SDD}$ can be rewrite as follows,

$$\mathcal{L}_{SDD} = \sum_{m \in M} \sum_{n \in N_m} \beta \mathcal{D}_{con}(m, n) + \gamma \mathcal{D}_{com}(m, n) \tag{10}$$

where $\mathcal{D}_{con}(m, n)$ and $\mathcal{D}_{com}(m, n)$ denotes the loss for consistent and complementary logit knowledge respectively.

Finally, combined with label supervision, the total training loss for the student to leverage multi-scale patterns from the teacher to improve its performance is defined as follows,

$$\mathcal{L}_1 = \mathcal{L}_{CE} + \alpha \mathcal{L}_{SDD} \tag{11}$$

where $\mathcal{L}_{CE}(.,.)$ denotes the label supervision loss for the task at hand, $\alpha$ is a balancing factor.

**Compared with conventional KD**: Particularly, we can derive that $P_T^g = \pi_T(m, n)$ and $P_S^g = \pi_S(m, n)$ when $m = 1$, $n = 1$. Thus, the conventional knowledge distillation loss can be regarded as a term of the SDD loss, covering the entire image $(0,0,w*d,w*d)$. It encourages the student to learn the contextual information of the whole image from the global logit output of the teacher. Besides, when $m > 1$, SDD further calculates the logit output at local scale loss to preserve the fine-grained semantic knowledge. These terms guide the student to inherit comprehensive knowledge from the teacher, enhancing its discrimination ability to the ambiguous samples.

**Compared with multi-branch KD**: Multi-branch KD, such as ONE (Zhu et al., 2018) and DCM (Yao & Sun, 2020), usually consists of multiple classifiers and fuses their logit output as the teacher knowledge to guide the training of the student. This introduces additional structure complexity. In contrast, although SDD introduces multi-scale pooling to generate multi-scale features, it calculates the multi-scale logit output via the same classifier. In other words, SDD is still the single-branch method and does not introduce any extra structural and computational complexity.

**Compared with DKD**: DKD (Zhao et al., 2022) decouples the logit knowledge as the target class and non-target class knowledge, which is conducted on the class scale and performed after calculating the logit output. In contrast, SDD decouples the logit knowledge as multi-scale knowledge, which is conducted on the spatial scale and performed before calculating the logit output. In particular, DKD can also be embedded in SDD and achieves better performance (see Table 2, Table 3).

## 4 EXPERIMENTS

We conduct experiments on the classical model compression (Cheng et al., 2017; Deng et al., 2020) task to evaluate the effectiveness of the proposed SDD method for the teacher-student pairs that have different network architectures, respectively.

| Teacher | ResNet32x4 | WRN40-2 | WRN40-2 | ResNet50 |
|---|---|---|---|---|
| | 79.42 | 75.61 | 75.61 | 79.34 |
| Student | MobileNetV2 | VGG8 | MobileNetV2 | ShuffleNetV1 |
| | 64.6 | 70.36 | 64.6 | 70.50 |
| FitNet | 65.61 | 70.98 | 65.12 | 72.03 |
| SP | 67.52 | 73.18 | 66.34 | 71.28 |
| CRD | 69.13 | 73.88 | 68.89 | 75.70 |
| SemCKD | 68.99 | 73.67 | 68.34 | 75.56 |
| ReviewKD | - | - | - | - |
| MGD | 68.13 | 73.33 | 68.55 | 74.99 |
| KD | 67.72 | 73.97 | 68.87 | 75.82 |
| SD-KD | 69.33(+**1.61**) | 74.44(+ **0.57**) | 69.91(+**1.04**) | 76.87(+**1.05**) |
| DKD | 68.98 | 74.33 | 69.33 | 77.01 |
| SD-DKD | 70.18(+**1.2**) | 74.88(+**0.55**) | 70.23(+**0.9**) | 78.11(+**1.1**) |
| NKD | 68.81 | 73.68 | 68.85 | 76.22 |
| SD-NKD | 69.69(+**0.88**) | 74.21(+**0.53**) | 69.84(+**0.99**) | 77.05(+**0.83**) |
| SSKD | 70.45 | 75.11 | 71.50 | 77.61 |
| SD-SSKD | **71.35**(+**0.9**) | **75.60**(+**0.49**) | **72.21**(+**0.71**) | **78.24**(+**0.63**) |

Table 1: Performance of model compression on the CIFAR-100 dataset. Here, the teacher and student with different network structures and layers. Specifically, the layers of ResNet32x4, WRN40_2, and ResNet50 are 4,4, and 3 while the layers of MobileNetV2, VGG8, and ShuffleNetV1 are 3,3, and 4, respectively.

## 4.1 EXPERIMENTAL SETUPS

**Datasets.** For the model compression task, we conduct the experiments on the CIFAR100 (Krizhevsky et al., 2009), CUB200 (Welinder et al., 2010), and ImageNet (Deng, 2009). Here, CIFAR100 and ImageNet are used for the evaluation of classic classification tasks. CUB200 is used for the evaluation of fine-grained classification tasks, which includes 200 different species of birds.

| Teacher | ResNet32x4 | WRN40-2 | ResNet50 | VGG13 | ResNet32x4 | ResNet50 |
|---|---|---|---|---|---|---|
| Acc | 79.42 | 75.61 | 79.34 | 74.64 | 79.42 | 79.34 |
| Student | ShufleNetV1 | ShufleNetV1 | MobileNetV2 | MobileNetV2 | ShuffleNetV2 | VGG8 |
| Acc | 70.50 | 70.50 | 64.6 | 64.6 | 71.82 | 70.36 |
| FitNet | 73.54 | 73.73 | 63.16 | 63.16 | 73.54 | 70.69 |
| RKD | 72.28 | 72.21 | 64.43 | 64.43 | 73.21 | 71.50 |
| SP | 73.48 | 74.52 | 68.08 | 68.08 | 74.56 | 73.34 |
| PKT | 74.10 | 72.21 | 66.52 | 66.52 | 74.69 | 73.01 |
| CRD | 75.11 | 76.05 | 69.11 | 69.11 | 75.65 | 74.30 |
| WCoRD | 75.77 | 76.32 | 70.45 | 69.47 | 75.96 | 74.86 |
| SemCKD | 76.31 | 76.06 | 68.69 | 69.98 | 77.02 | 74.18 |
| ReviewKD | 77.45 | 77.14 | 69.89 | 70.37 | 77.78 | 75.34 |
| MGD | 76.22 | 75.89 | 68.54 | 69.44 | 76.65 | 73.89 |
| KD | 74.07 | 74.83 | 67.35 | 67.37 | 74.45 | 73.81 |
| SD-KD | 76.30(+**2.23**) | 76.85(+**2.02**) | 69.95(+**2.60**) | 68.99(+**1.62**) | 77.47(+**3.02**) | 74.89(+**1.08**) |
| DKD | 76.45 | 76.70 | 70.35 | 69.71 | 77.07 | 75.34 |
| SD-DKD | 77.50(+**1.05**) | 77.21(+**0.51**) | 71.36(+**1.01**) | 70.55(+**0.84**) | 78.05(+**0.98**) | 75.86 (+**0.52**) |
| NKD | 75.51 | 75.96 | 69.39 | 68.72 | 76.26 | 74.01 |
| SD-NKD | 76.64(+**1.12**) | 76.91(+**0.95**) | 70.05(+**0.66**) | 70.10(+**1.38**) | 76.97(+**0.71**) | 74.52(+**0.51**) |

Table 2: Performance of model compression on the CIFAR-100 dataset. Here, the teacher and student with different network structures but the same layer.

|       | Teacher | Student | KD    | **SD-KD**      | DKD   | **SD-DKD**     | NKD   | **SD-NKD**     |
|-------|---------|---------|-------|----------------|-------|----------------|-------|----------------|
| Top1  | 76.16   | 68.87   | 70.50 | 71.24(**+0.74**) | 72.05 | 72.58(**+0.53**) | 72.58 | 73.12(**+0.54**) |
| Top5  | 92.86   | 88.76   | 90.34 | 90.41          | 91.00 | 90.71          | 90.80 | 91.11          |

Table 3: Top-1 and top-5 accuracy (%) on the ImageNet validation. We set ResNet-50 as the teacher and MobileNet-V1 as the student.

**Implementation Details.** As described in Section 3.2, the $\mathcal{LD}$ used for $L_{SDD}$ can be implemented with arbitrary logit-based loss. Here, for a fair comparison with previous logit-based methods and to study the effectiveness of the SDD, we use the same losses used in the KD, DKD, and NKD methods and denote these implementations as **SD-KD SD-DKD**, and**SD-NKD**, respectively.

SDD has four hyper-parameters, scale set $M$, balance parameters $\alpha$, $\beta$ and $\gamma$. The detailed analysis of $M$,$\beta$ and $\gamma$ can be seen in the appendix. Specifically, $M$ is set as $\{1, 2, 4\}$ for the distillation tasks with input or structural differences, and $\{1, 2\}$ for the distillation tasks with similar structures. $\beta$ and $\gamma$ are set as 1.0 and 2.0. As for the $\alpha$, for a fair comparison with previous logit-based methods, we follow the same setting used in the original KD, DKD, and NKD methods, namely $\alpha = 1$. In particular, since the sum of multi-scale logit distillation loss could be high and lead to a large initial loss, we utilize a 30-epoch linear warmup for all experiments. Finally, the training settings for different datasets and teacher and student pairs are attached in the appendix due to the page limit.

## 4.2 COMPARISON RESULTS

**Results on the teacher and student with different network structures and layers.** As shown in Table 1, SDD contributes to significant performance gains for multiple classical logit distillation methods. Specifically, SDD increases the performance of ResNet32x4-MobileNetV2, WRN40-2-MobileNetV2, and ResNet50-ShuffleNetV1 by nearly 1% for KD, DKD, and NKD. These results show the effectiveness of the proposed SDD method in dealing with the teacher and student with different network structures and layers. Moreover, even the conventional KD outperforms the state-of-the-art feature-based methods in some teacher and student pairs, such as WRN_2 and VGG8 as well as ResNet50 and ShufleNetV1. These results show the superiority of logit distillation and the necessity of improving logit knowledge distillation. Here, the result of the ReviewKD method is $-$ since it cannot handle such scenarios.

**Results on the teacher and student with different network structures but the same layer.** As shown in Table 2 and Table 3, SDD improves the performance of multiple classical logit distillation methods by 0.5%~2.23%. Specifically, for most teacher-student pairs, SDD contributes to more than 1% gains, demonstrating its effectiveness to deal with the teacher and student with different network structures but the same layer. Besides, the proposed SD-DKD outperforms all the feature-based distillation methods, including the state-of-the-art method ReviewKD and MGD. This further confirms the superiority of SDD.

**Results on fine-grained classification tasks.** Results in Table 4 demonstrate that our SDD achieves more remarkable performance gains on such a challenging task. Specifically, SDD improves the performance of multiple classical logit distillation methods by 1.06%~6.41%. The reason may be that fine-grained classification tasks have a stronger dependence on fine-grained semantic information than the conventional classification task in CIFAR100. This demonstrates the potential of the proposed SDD for distilling fine-grained classification models.

## 4.3 ABLATION STUDY

The ablation studies are conducted on CIFAR-100 by using the heterogeneous teacher-student pair ResNet32x4/ShuffleNetV2 and homogeneous pair ResNet32x4/ResNet8x4, respectively.

**Training efficiency.** We assess the training time of state-of-the-art distillation methods to evaluate the efficiency of SDD. As shown in Table 9, the training time of SD-KD is same to the KD and less than the feature-based methods. This is because SDD calculates the multi-scale logit output via the same classifier, introducing no extra structural complexity.

| Teacher | ResNet32x4 | ResNet32x4 | VGG13 | VGG13 | ResNet50 |
|---|---|---|---|---|---|
| Acc | 66.17 | 66.17 | 70.19 | 70.19 | 60.01 |
| Student | MobileNetV2 | ShuffleNetV1 | MobileNetV2 | VGG8 | ShuffleNetV1 |
| Acc | 40.23 | 37.28 | 40.23 | 46.32 | 37.28 |
| SP | 48.49 | 61.83 | 44.28 | 54.78 | 55.31 |
| CRD | 57.45 | 62.28 | 56.45 | 66.10 | 57.45 |
| SemCKD | 56.89 | 63.78 | 68.23 | 66.54 | 57.20 |
| ReviewKD | - | 64.12 | 58.66 | 67.10 | - |
| MGD | - | - | - | 66.89 | 57.12 |
| KD | 56.09 | 61.68 | 53.98 | 64.18 | 57.21 |
| SD-KD | 60.51(**+4.42**) | 65.46(**+3.78**) | 59.80(**+5.82**) | 67.32(**+3.14**) | 60.56(**+3.25**) |
| DKD | 59.94 | 64.51 | 58.45 | 67.20 | 59.21 |
| SD-DKD | 62.97(**+3.43**) | 65.58(**+1.06**) | 64.86(**+6.41**) | 68.67(**+1.47**) | 60.66(**+1.45**) |
| NKD | 59.81 | 64.0 | 58.40 | 67.16 | 59.11 |
| SD-NKD | 62.69(**+2.88**) | 65.50(**+1.5**) | 64.63(**+6.23**) | 68.37(**+1.21**) | 60.42(**+1.31**) |

Table 4: Performance of model compression on the CUB200 dataset. Here, we conduct experiments on three different teacher-student pairs, the same structure and layer (VGG13-VGG8), different structures while the same layer (ResNet32-ShffleNetV1 and VGG13-MobileNetV2), and different structures and layers (ResNet32x4-MobileNetV2 and ResNet50-ShuffleNetV1).

| Methods | CRD | ReviewKD | KD | SD-KD |
|---|---|---|---|---|
| Times(ms) | 41 | 26 | 11 | 11 |

Table 5: Training time (per batch) vs. accuracy on CIFAR-100. We set ResNet32×4 as the teacher and ResNet8×4 as the student.

**Effect of the decoupled logit knowledge**. As shown in Table 6, the fusion of consistent and complementary logit knowledge improves the performance of conventional KD by 3.30% and 2.23% for homogeneous and heterogeneous teacher-student pairs, respectively. This demonstrates the effectiveness of decoupling the global logit knowledge into multiple semantic-independent logit knowledge. Besides both consistent and complementary logit knowledge consistently improve the performance of conventional KD for homogeneous and heterogeneous teacher-student pairs, respectively, verifying their effectiveness.

**Visualizations.** We present visualizations from two perspectives (with setting teacher as ResNet32x4 and the student as ResNet8x4 on CIFAR-100). (1) Fig. 3 shows the difference of correlation matrices of the global logits of student and teacher. Different from the DKD method, SD-KD provides a similar difference map as the KD. This indicates that the improvement of SDD does not come from better imitating the global logit output of the teacher. To further study the mechanism of SDD, we further visualize some cases that can be classified correctly by the student trained with SD-KD while misclassified by the student trained with conventional KD (Fig. 4). From the figure, we can see that the samples misclassified by the KD model are exactly the ambiguous samples that seem similar in the global semantics. This verifies our proposition that SDD can help the student acquire

| | ResNet32x4-ResNet8x4 | ResNet32x4-ShuffleNetV1 |
|---|---|---|
| N/A | 73.33 | 74.07 |
| Consistent | 75.14 | 75.88 |
| Complementary | 74.79 | 75.10 |
| Fusion | **76.63** | **76.30** |

Table 6: Performance of the SDD method that uses different decoupled logit knowledge on the teacher-student pairs with homogeneous and heterogeneous network structures. The $\mathcal{LD}$ used for $L_{SDD}$ is conventional logit KD loss.

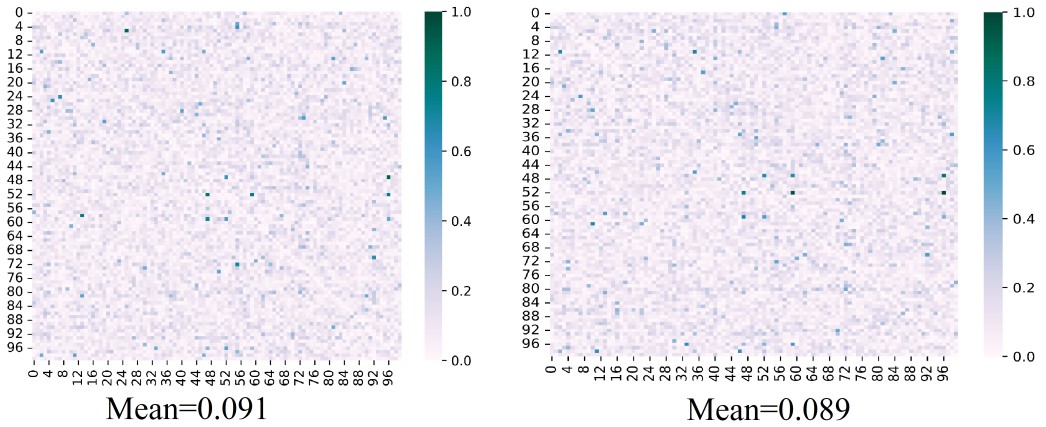

Mean=0.091        Mean=0.089

Figure 3: Difference of correlation matrices of student and teacher logits of KD (left) and SD-KD (right).

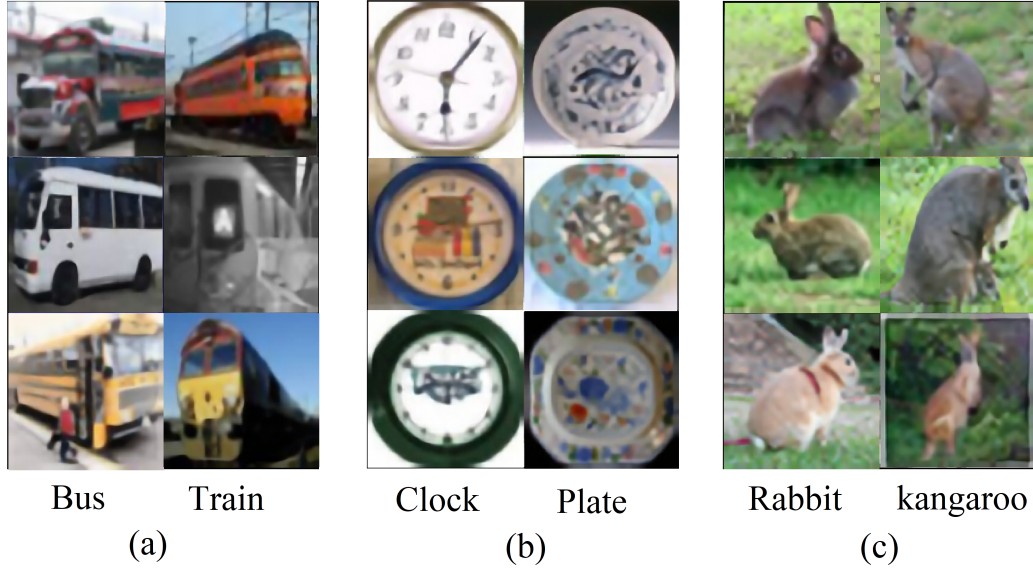

Bus    Train       Clock    Plate       Rabbit    kangaroo

(a)           (b)           (c)

Figure 4: Some examples that can be classified correctly by the student trained with SD-KD while misclassified by the student trained with conventional KD.

the fine-grained semantic information of local regions to regularize global knowledge, preventing it from overfitting the ambiguous samples.

## 5 CONCLUSION

**Conclusion**.This paper revisits conventional logit-based distillation and reveals that it is the coupled semantic knowledge in global logit knowledge that limits its effectiveness. To address this issue, we propose the semantic decoupled knowledge distillation to decouple the global logit output as consistent and complementary local logit output and establish the knowledge-transferring pipeline for them to mine and transfer richer and fine-grained semantic knowledge. Finally, extensive experiments demonstrate the effectiveness of SDD for wide teacher-student pairs, specifically for the fine-grained classification task.

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

# 6 APPENDIX

## 6.1 EFFECT OF DIFFERENT DECOUPLED SCALES ?

We set different scale sets $M$ for the $L_{SDD}$ to study the effect of local logit knowledge at different scales. Specifically, each scale $m$ in $M$ means the logit knowledge of region $\mathcal{Z}(m, n)$ is utilized. For a fair comparison, we take the model only using the global knowledge as the baseline, namely $M = \{1\}$. The results are shown in Table 7.

For the teacher-student pair of ResNet32x4 and ShuffleNetV2, SDD achieves the best results when $M = \{1, 2, 4\}$. In contrast, the teacher-student pair of ResNet32x4 and ResNet8x4 reaches the best results when $M = \{1, 2\}$. This indicates that the teacher-student pair with heterogeneous structures needs more fine-grained semantic knowledge than the teacher-student pair with homogeneous structures. This can be explained by knowledge regularization. Specifically, when the global knowledge is not valid, such as suffering from ambiguous samples, the local knowledge can make a difference. Here, the structure heterogeneity is not conducive for students to imitate the teacher's knowledge. Thus the student needs more local logit knowledge to assist its training. Therefore, scale set $M$ of SD-KD and SD-DKD is set as $\{1, 2, 4\}$ for the distillation tasks with input or structural differences, and $\{1, 2\}$ for the distillation tasks with similar structures.

|  | Resnet32-ShuffleNetV2 | ResNet32x4-ResNet8x4 |
|---|---|---|
| M={1} | 74.07 | 73.33 |
| M={1,2} | 75.10 | **76.63** |
| M={1,4} | 75.84 | 76.13 |
| m={1,2,4} | **76.30** | 75.74 |

Table 7: Performance of the SDD method that uses local logit knowledge at different scales on the teacher-student pairs with heterogeneous and homogeneous network structures. The $\mathcal{LD}$ used for $L_{SDD}$ is conventional logit KD loss.

## 6.2 HYPER-PARAMETER ANALYSIS

. The results are shown in Table 8. We first fix $\gamma$ as 1.0 in the first row for simplification and get the optimal $\beta$ as 2.0. Then we fix $\beta$ as 2.0 in the second row since it achieves the best performance in the first. As a result, we can get the optimal combination of $(\alpha, \beta)$ as $(1.0, 2.0)$.

| r | 1 | 2 | 4 | 6 | 8 |
|---|---|---|---|---|---|
| ACC | 76.22 | **76.30** | 75.99 | 76.15 | 75.78 |

| b | 0.2 | 0.5 | 0.8 | 1.0 | 1.5 |
|---|---|---|---|---|---|
| ACC | 75.33 | 76.22 | 75.92 | **76.30** | 75.82 |

Table 8: Perform of SDD method with different $\beta$ and $\gamma$ on ResNet32×4 and ShuffleV1.

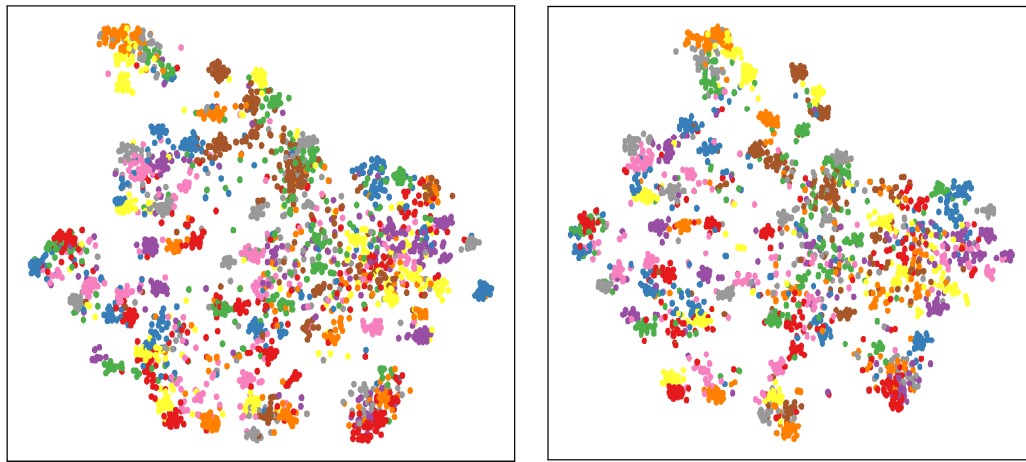

Figure 5: t-SNE of features learned by KD (left) and SD-KD (right).

### 6.3 TRAINING DETAILS

For the model compression tasks, we follow the same setting used in the CRD model to process data and train the models, which is widely used in recent distillation studies, such as SemCKD (Chen et al., 2021a) and DKD (Zhao et al., 2022). For the privilege-based action recognition task, we split the NTUD60 dataset by the cross-subject protocol provided in the original paper, using twenty individuals for training and the remaining for testing (Shahroudy et al., 2016). For fair comparisons, we take the classic I3D (Carreira & Zisserman, 2017) as the backbone of teacher and student networks, which is trained with the SGD optimizer, where the learning rate is 0.001, momentum is 0.9, the batch size is 16, and the maximum epoch is 250.

**Training efficiency.** We assess the training time of state-of-the-art distillation methods to evaluate the efficiency of SDD. As shown in Table 9, the training time of SD-KD is similar to the KD and less than the feature-based methods. Since SDD is reformulated from the classical KD, it needs almost the same computational complexity as KD, and of course no extra parameters. However, feature-based distillation methods require extra training time for distilling intermediate layer features.

| Methods | CRD | ReviewKD | KD | SD-KD |
|---|---|---|---|---|
| Times(ms) | 41 | 26 | 11 | 11 |

Table 9: Training time (per batch) vs. accuracy on CIFAR-100. We set ResNet32×4 as the teacher and ResNet8×4 as the student.

