# OpenReview forum: "Semantic Decoupled Distillation"
_ICLR.cc/2024/Conference — ICLR 2024 Conference Withdrawn Submission_

### Official Review · Reviewer_SyKS · 2023-10-30

**Soundness:** 3 good
**Presentation:** 3 good
**Contribution:** 2 fair
**Rating:** 3
**Confidence:** 3

**Summary:**

This paper proposed a method, semantic decoupled distillation (SDD), to improve logit distillation. The authors found that traditional logit distillation coupled multiple semantic knowledge in a single logit output, resulting in sub-optimal performance. The SDD method decoupled the whole logit output into the logit outputs of multiple local regions which acquired unambiguous semantic knowledge. SDD divided these logit outputs into consistent and complementary terms based on their class. Additionally, SSD used dynamic weights to adapt to different tasks and data scenes.

Contributions:
1.	The authors indicated the weakness of current methods that hindered the student from inheriting comprehensive knowledge from the teacher.
2.	The authors proposed SDD to assist the logit distillation and proved the effectiveness of their method on several benchmark datasets.

**Strengths:**

The rationale for enhancing the classic distillation is well-founded, and their explanation of the distinction between M in heterogeneous and homogeneous structures in Appendix 6.1 is robust.
The visualization part affirmed that the improvement of SDD did not arise from a more accurate imitation of the global logit output of the teacher. The case study further verified their proposition.

**Weaknesses:**

In the introduction section, the authors mentioned that SDD introduced dynamic weights for consistency and complementary to adapt to different tasks and data scenes, while the rest of the paper only set them to fixed numbers.
The authors forget to mention and compare to many recent knowledge distillation methods. their results are below the state-of-art results. for example, using resnet 32X4 as a teacher and Shuffle V2 as a student, current state-of-art results can reach 79.54 [1].

[1] Ding, Fei, et al. "Dual-Level Knowledge Distillation via Knowledge Alignment and Correlation." IEEE Transactions on Neural Networks and Learning Systems

**Questions:**

In Appendix 6.2, did you set the balance parameter α to 1? If α is 1, why did you adjust γ  and β separately instead of utilizing a single balance parameter?

---

### Official Review · Reviewer_q6qm · 2023-11-01

**Soundness:** 2 fair
**Presentation:** 2 fair
**Contribution:** 2 fair
**Rating:** 5
**Confidence:** 5

**Summary:**

This paper focused on logit knowledge distillation. The authors argued that existing logit-based methods only leverage the global logit output, which may hinder the student from learning comprehensive knowledge from the teacher. As such, they proposed a semantic decoupled distillation method (SSD), decoupling the logit output of the whole image into the logit outputs of multiple local regions.

**Strengths:**

1. The authors conducted extensive experiments to evaluate the effectiveness of SSD.
2. Knowledge distillation is a hot-topic and attracts wide interest in the research community.

**Weaknesses:**

1. In the second and third paragraphs of the introduction, the connection between global logit knowledge and the coupled information of multiple classes is not very clear.
2. This paper should give intuition, theoretical analysis, or evidence to explain why the logit output with coupled information will hinder the learning of comprehensive knowledge in the student model. In most situations, the logit output from a teacher model essentially provides the knowledge among multiple classes (see the discussion in DKD).
3. The derived logit output from multi-scale pooling seems to still aggregate the information of a whole image by average pooling. The contribution for knowledge distillation itself is limited.
4. To perform distillation in multiple scales, their distillation losses will increase training cost (time and memory). It is better to discuss their complexity and effects.

**Questions:**

1. In the notation, some definitions are confusing for readers. For example, $f_{Net}(j,k)$ is the feature vector at the location $(i,j)$. Where is the $i$ in the $f_{Net}(j,k)$? In the first paragraph of Section 3.2, where is $L_T$ in Eq. 5. What is the difference between $P_t$ in Eq. 2 and $P_T$ in Eq. 4?
2. The technical details of dividing consistent and complementary logit are missing. How do you perform this operation? It is better to give some formal definitions.

---

### Official Review · Reviewer_PAHr · 2023-11-01

**Soundness:** 3 good
**Presentation:** 3 good
**Contribution:** 2 fair
**Rating:** 5
**Confidence:** 5

**Summary:**

The paper proposes a kind of logit distillation method: semantic decoupled distillation, which instead uses multiple local outputs to produce local logit, transferring richer knowledge to student models.  Extensive experiments demonstrate SDD's effectiveness, especially in fine-grained classification with diverse teacher-student pairs.

**Strengths:**

1. Compared with baseline methods, the results of the proposed method are good.
2. The paper is well organized and the writing is clear.

**Weaknesses:**

1. The novelty is limited. The method is named 'semantic decoupling distillation', but actually using multi-scale pooling to replace the global pooling, which seems nothing to do with the "semantic decoupling".  The loss for consistent and complementary logit knowledge is simply using different weight to sum up, but the weight is set as hyperparameters which is not dynamic.
2. The results are mostly using CNN models and based on small scale dataset (CIFAR 100). To validate the method's effectiveness, more models (such as vision transformers) and ablation on bigger dataset are preferred.

**Questions:**

The same as listed in weaknesses.

---

### Official Review · Reviewer_onwW · 2023-11-06

**Soundness:** 2 fair
**Presentation:** 1 poor
**Contribution:** 2 fair
**Rating:** 3
**Confidence:** 2

**Summary:**

The paper introduces the idea of using multi-scale pooling to generate multi-scale features to use in knowledge distillation. This is in contrast to the logit-based approach to distillation which just uses the output of the average pooling). This "decouples the logit output of the whole input into the logit outputs of multiple local regions". The authors note that this approach works well especially in fine-grained classification tasks.

**Strengths:**

The biggest strength of the paper is the general ideal; doing pooling at multiple scales and using those values as what to compare the student/teacher models for the purpose of loss makes a lot of sense. It clearly shows how more information can be transferred between the student and the teacher. The authors demonstrate how the models can have different structures and the approach still works. They did an expansive evaluation comparing it to other distillation approaches.

**Weaknesses:**

In general, the paper is poorly written. There is inconsistency in how the figures are referenced. There are uncompleted sentences. But mostly, the paper is written with unnecessary complexity.

**Questions:**

- Please explain L_T and L_S, I was confused by this, why wouldn't it just be the softmax W_T, W_S?